# Interactive Query Answering on Knowledge Graphs with Soft Entity Constraints

**Daniel Daza**                                                                       *d.dazacruz@vu.nl*
*Translational AI Laboratory, Department of Laboratory Medicine*
*Amsterdam University Medical Center, Vrije Universiteit Amsterdam*

**Alberto Bernardi**                                                      *alberto.bernardi@accenture.com*
*Accenture Labs*

**Luca Costabello**                                                        *luca.costabello@accenture.com*
*Accenture Labs*

**Christophe Gueret**                                                  *christophe.gueret@accenture.com*
*Accenture Labs*

**Masoud Mansoury**                                                            *m.mansoury@tudelft.nl*
*Delft University of Technology*

**Michael Cochez**                                                             *michael.cochez@abo.fi*
*ELLIS Institute Finland & Abo Akademi University, Turku, Finland & Elsevier Discovery Lab, Amsterdam*

**Martijn Schut**                                                        *m.schut@amsterdamumc.nl*
*Translational AI Laboratory, Department of Laboratory Medicine*
*Amsterdam University Medical Center, Vrije Universiteit Amsterdam*

**Reviewed on OpenReview:** <https://openreview.net/forum?id=Qb6vIM7MxE>

## Abstract

Methods for query answering over incomplete knowledge graphs retrieve entities that are *likely* to be answers, which is particularly useful when such answers cannot be reached by direct graph traversal due to missing edges. However, existing approaches have focused on queries formalized using first-order-logic. In practice, many real-world queries involve constraints that are inherently vague or context-dependent, such as preferences for attributes or related categories. Addressing this gap, we introduce the problem of query answering with *soft constraints*. We formalize the problem and introduce two efficient methods designed to adjust query answer scores by incorporating soft constraints without disrupting the original answers to a query. These methods are lightweight, requiring tuning only two hyperparameters or a small neural network trained to capture soft constraints while maintaining the original ranking structure. To evaluate the task, we extend existing QA benchmarks by generating datasets with soft constraints. Our experiments demonstrate that our methods can capture soft constraints while maintaining robust query answering performance and adding very little overhead. With our work, we explore a new and flexible way to interact with graph databases that allows users to specify their preferences by providing examples interactively.

Our code is available at <https://github.com/dfdazac/nqr>.

## 1 Introduction

Knowledge graphs (KGs) are graph-structured databases that store information about a domain using triples of the form *(subject, predicate, object)*. Such a compact representation enables efficient algorithms for

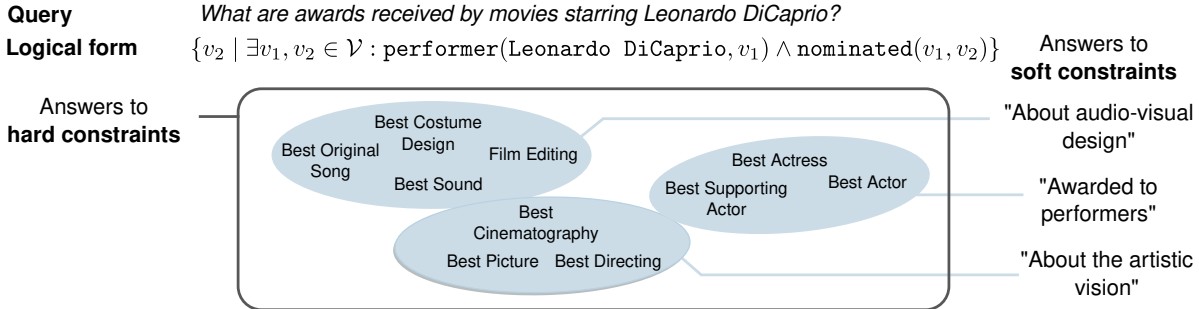

Figure 1: A query over a KG can be written in a logical form that specifies *hard constraints* that an entity must meet. We investigate the problem of incorporating *soft constraints* that cannot be specified in logical form, such as "*awards given to performers.*" Such constraints correspond to subsets of entities, shown in shaded clusters, that share some similarities.

answering queries about entities over the domain encoded by the KG (Fensel et al., 2020; Hogan et al., 2021). In principle, answers to these queries can be obtained by traversing the graph and collecting the entities that satisfy the given conditions. However, this strategy misses answers due to connections that are not explicit in a KG that may be incomplete, but that are *likely* to satisfy a query, for example because of their similarity to known answers. The problem of finding likely answers to simple queries over KGs has been studied in areas such as rule mining (Galárraga et al., 2013; Meilicke et al., 2018; 2019; Qu et al., 2020) and link prediction (Nickel et al., 2016; Ji et al., 2022), and complex query answering (Hamilton et al., 2018; Daza & Cochez, 2020; Ren et al., 2020; Ren & Leskovec, 2020; Arakelyan et al., 2021; 2023; Bai et al., 2023; Zhu et al., 2022; Ren et al., 2024). Given a query, these methods score all the entities in the KG, indicating the likelihood of meeting all conditions specified in the query. Notably, prior works have focused on conditions that can be specified using different subsets of first order logic.

We identify a relevant and complementary problem: answering queries with constraints that cannot be specified formally as further conditions in a query. We refer to these as *soft constraints*, to contrast them with the *hard* constraints that can be specified using first order logic. Consider the following query: "*What are award nominations received by movies starring Leonardo DiCaprio?*". A user might be additionally interested in nominations "*related to the audio-visual design of the movie*", such as *Best Costume Design*, or *Best Sound*, and not in nominations about the artistic vision of a film (see Fig. 1). Such a constraint cannot be specified using the language of first order logic, simply because such an idea of relatedness is too vague to be formalized into precise logical statements. Our goal is thus to broaden the expressivity of queries on KGs which are specified using first order logic, by incorporating additional soft constraints to refine the set of answers.

The challenge of incorporating soft constraints lies in adjusting answer scores with minimal deviation from the original distribution of scores, so that hard logical constraints imposed by the query are still met. Training a model for this task requires datasets that capture the relationships between query answers and soft constraints – data that is not available in existing query answering benchmarks over incomplete KGs. Additionally, the reranking process must be efficient and integrate seamlessly with state-of-the-art QA models.

To tackle these challenges, we formalize the problem and introduce efficient methods designed to capture soft constraints while preserving the global structure of the original list of scores for a given query. In summary, our work makes the following key contributions:

1. We introduce the problem of query answering on knowledge graphs with soft entity constraints, defining it as an interactive process where preferences are specified incrementally.

2. We introduce two computationally efficient methods for reranking query answers based on an initial list of scores from an existing query answering system. The methods are lightweight, requiring tuning two parameters or a small neural network optimized to capture soft constraints while maintaining the original ranking structure.

3. We extend existing benchmarks for query answering with automatically generated soft constraints comprising a large and diverse set of queries with different notions of similarity. Through comprehensive evaluations, we show how our methods effectively capture these constraints without significantly compromising global ranking performance or runtime performance.

Our results point toward a more flexible interaction paradigm that complements existing methods for query answering on graphs, where answers can be adjusted according to preferences specified interactively.

## 2 Related Work

**Approximate Query Answering.**

Recent works have introduced methods for inferring plausible answers that are not reachable via explicit graph traversal, typically due to incompleteness in the graph. We build on methods for learning on KGs that retrieve query answers that are not reachable via graph traversal. Early work focused on simple link prediction, where an edge is predicted between two entities (Nickel et al., 2011; Bordes et al., 2013; Trouillon et al., 2016; Lacroix et al., 2018; Sun et al., 2019). More recently, the set of queries that learning-based methods can tackle has been broadened to complex queries (e.g., paths of length two or three with intermediate variables), using a formulation of queries based on a small (Hamilton et al., 2018; Daza & Cochez, 2020; Ren et al., 2020; Ren & Leskovec, 2020; Arakelyan et al., 2021; 2023; Bai et al., 2023; Zhu et al., 2022; Ren et al., 2024) or a larger (Cucumides et al., 2024; Yin et al., 2024) subset of first order logic. In all these cases, the output of a query answering model is a score for each entity in the graph that indicates how likely it is to meet constraints specified using first order logic. Our work complements these approaches by supporting soft constraints that cannot be expressed in the same way, due to the inherent limitations of the language of first order logic.

**Active Learning on KGs.** The idea of collecting entities that exemplify soft constraints is closely related to the problem of active learning (Fu et al., 2013; Ren et al., 2022). Active learning allows training models incrementally when data is of limited availability. In the context of KGs, it has been applied to sampling training data for link prediction models under a constrained computational budget (Ostapuk et al., 2019), error detection (Dong et al., 2023), and KG alignment (Huang et al., 2023); all cases in which a prediction is well-defined. Applying active learning directly to an existing query answering model in order to capture soft constraints is not trivial, as a single query can have different types of soft constraints (as illustrated in Fig. 1), potentially leading to contradicting model updates. We avoid this by training a lightweight neural query reranker via supervised learning, using automatically generated data of soft preferences.

**Information Retrieval (IR) and Recommender Systems.** A common task in IR is to retrieve documents relevant to a given query (Manning et al., 2008), for example based on their similarity to a textual query using sequence embeddings (Devlin et al., 2019). We adopt a similar approach in one of the baselines in our experiments. We also draw inspiration from Learning to Rank (LTR), where models are trained to order items according to their relative relevance. Pairwise methods such as RankNet and LambdaRank (Burges et al., 2005; 2006) and their boosted extension LambdaMART (Burges, 2010) learn from preference pairs by directly optimizing ranking metrics such as NDCG, and have been shown to outperform regression- or classification-based ranking approaches. Modern implementations, such as the one provided in LightGBM (Ke et al., 2017), efficiently extend these methods to large datasets while supporting graded relevance labels and fast tree-based inference. These approaches generate a new ranking from scratch given a set of preferences, while we aim to preserve original score relationships while biasing the ranking towards soft constraints.

Other learning-to-rank methods take a list-wise approach that considers permutations of candidate lists (Cao et al., 2007; Xia et al., 2008; Zhu & Klabjan, 2020), but these approaches do not scale well to the number of entities typically found in a knowledge graph.

Lastly, the field of Recommender Systems has explored the use of interaction data (e.g., clicks, ratings, and browsing history) to generate recommendations (Koren et al., 2021). Various collaborative filtering models exist in the literature (Ning & Karypis, 2011; Steck & Liang, 2021; Koren et al., 2009; Rendle et al., 2012; He et al., 2017; Christoffel et al., 2015), some of which demonstrate the effectiveness of neural networks at capturing preference data. Other approaches, such as Rocchio's algorithm (Rocchio Jr, 1971; Manning et al.,

2008), introduce feedback into queries by updating a query vector with a new one as examples are added. However, our problem differs from this setting as the starting point consists of the scores of a base QA model for the unconstrained query, for which no query vectors are available that can be updated.

## 3 Preliminaries

We denote a KG as a tuple $\mathcal{G} = (\mathcal{V}, \mathcal{E}, \mathcal{R})$, where $\mathcal{V}$ is the set of entities, $\mathcal{R}$ is the set of relations, and $\mathcal{E}$ is the set of edges of the form $(h, r, t)$ where $h, t \in \mathcal{V}$ are the head and tail entities, and $r \in \mathcal{R}$ is the relation between them. A triples $(h, r, t)$ in a KG implies the truth of the statement $r(h, t)$ in first-order logic, where $r \in \mathcal{R}$ is a binary predicate, and $h, t \in \mathcal{V}$ its arguments.

**Logical queries over KGs.** A query $q$ over $\mathcal{G}$ defines a set of constraints that an entity must satisfy to be considered an answer. For example, consider the query *What are award nominations received by movies starring Leonardo DiCaprio?* The *target* variable of this query is an award, and the query places two constraints on it: 1) the award nomination is given to movies, and 2) the movie features Leonardo DiCaprio. Formally, we can express the set $A_q$ of answers to the query using first order logic as follows:

$$A_q = \{v_2 \mid \exists v_1, v_2 \in \mathcal{V} : \texttt{performer}(\texttt{Leonardo DiCaprio}, v_1) \wedge \texttt{nominated}(v_1, v_2)\}. \tag{1}$$

Answering this query requires assigning entities from $\mathcal{V}$ to variables $v_1$ and $v_2$ such that the conjunction specifying the constraints is true according to the information contained in the KG. More generally, and following the notation in Bai et al. (2023), we consider first-order logic (FOL) queries that can include conjunctions, disjunctions, and negations, which we write in disjunctive normal form as follows:

$$\mathcal{A}_q = \{v_t \mid \exists v_1, \ldots, v_N \in \mathcal{V} : (c_1^1 \wedge \cdots \wedge c_{m_1}^1) \vee \cdots \vee (c_1^n \wedge \cdots \wedge c_{m_n}^n)\}, \tag{2}$$

where $v_1, \ldots, v_N$ are variables in the query, and $v_t$ is the *target* variable. The constraints $c_j^i$ of the query consist of binary relations between two variables, or between a variable and a known entity in $\mathcal{V}$, while optionally including negations:

$$c_j^i = \begin{cases} r(e, v) \text{ or } r(v', v) \\ \neg r(e, v) \text{ or } \neg r(v', v) \end{cases} \quad \text{with } e \in \mathcal{V}, v, v' \in \{v_t, v_1, \ldots, v_N\}, \text{ and } r \in \mathcal{R}.$$

**Approximate Query Answering.** In order to provide answers to queries that are not limited to the information explicitly stated by the edges in the graph, recent works have introduced machine learning models for *approximate* query answering (QA) (Hamilton et al., 2018; Daza & Cochez, 2020; Ren et al., 2020; Ren & Leskovec, 2020; Arakelyan et al., 2021; 2023; Bai et al., 2023; Zhu et al., 2022; Ren et al., 2024; Cucumides et al., 2024; Yin et al., 2024). For a given query, these methods compute a score for all the entities in the KG according to how likely they are to be answers to a query. While the specific mechanisms for achieving this vary among methods, in general we denote a **QA model** as a function $a$ that maps a query $q$ to a vector $a(q) \in \mathbb{R}^{|\mathcal{V}|}$ of real-valued scores, with one score for each entity in the KG.

## 4 Query Answering with Soft Constraints

FOL queries over KGs allow imposing a broad range of constraints over entities, but this approach is still limited by the expressivity of first-order logic and the schema of the KG. In some applications one might require incorporating notions of similarity, which can be hard or impossible to express using FOL. We address this limitation by introducing the concept of *soft constraints*.

**Definition 1** (Soft constraint). *A **soft constraint** is a boolean predicate $s : \mathcal{V} \to \{0, 1\}$ that indicates whether an entity satisfies a desired semantic property. Furthermore, the predicate cannot be expressed using the logical form of Equation (2) and is therefore not directly evaluable from the triples in $\mathcal{E}$.*

Soft constraints can be used to further restrict the answer set of a query by requiring that the entity assigned to the target variable satisfies the constraint. Let $s : \mathcal{V} \to \{0, 1\}$ be a soft constraint predicate. The **constrained answer set** of query $q$ is defined as $\hat{A}_q = \{e \in \mathcal{A}_q \mid s(e)\}$.

**Preference sets.** Since the predicate $s$ is not present in the KG and it is not explicitly known, we propose to rely on a small set of labeled examples that can be provided *interactively* at different time steps $t$. Formally, a preference set is denoted as $P(t) = \{(e_1, s(e_1)), \ldots, (e_k, s(e_k))\}$, with $e_i \in \mathcal{V}$.

**Definition 2** (Interactive query answering with soft constraints). *Let $q$ be a query with hard logical constraints evaluated over a knowledge graph, $a^{(0)}(q)$ denote the initial list of scores produced by a query answering model, and $P(t)$ a preference set representing a soft constraint $s$. The problem of **interactive query answering with soft constraints** consists of computing an adjusted list of scores $a^{(t)}(q)$ that incorporates the information contained in $P(t)$.*

The adjusted scores should satisfy two desirable properties.

**Preserving of logical constraints.** Entities that do not satisfy the logical query should not be ranked above valid answers: $a^{(t)}(q)[u] > a^{(t)}(q)[v] \quad \forall u \in \mathcal{A}_q, \, \forall v \notin \mathcal{A}_q$.

**Capturing preferences.** Among entities satisfying the hard constraints, those that also satisfy the soft constraint should be ranked higher: $a^{(t)}(q)[u] > a^{(t)}(q)[v] \quad \forall u \in \hat{\mathcal{A}}_q, \, \forall v \in \mathcal{A}_q \setminus \hat{\mathcal{A}}_q$.

This formulation enables incorporating soft constraints dynamically, guiding the ranking of answers toward entities semantically aligned with the provided examples. Consider the query "*What are award nominations received by movies starring Leonardo DiCaprio?*" The corresponding logical form retrieves all entities $v_2$ such that there exists a movie $v_1$ with performer(Leonardo DiCaprio, $v_1$) and nominated($v_1, v_2$). Answers to the hard constraints include awards such as *Best Actor*, *Best Picture*, and *Best Sound*. In the case where we are interested in nominations *"related to the audio-visual design of the movie"*, but not those *"about the artistic vision"*, we can express this through the preference set $P(2) = \{(\text{Best Sound}, 1), (\text{Best Picture}, 0)\}$.

Here, the label $l = 1$ marks Best Sound as a positive example (entities similar to it, such as Best Original Song, should receive higher scores) while $l = 0$ marks Best Picture as a negative example, encouraging the model to de-emphasize similar entities (e.g., Best Director). After incorporating $P(2)$, the updated scores $a^{(2)}(q)$ should give priority to entities that reflect the soft constraint.

## 5 Incorporating Soft Constraints

We now turn our attention to the problem of adapting answer scores based on preference sets that represent a soft constraint. The goal is to develop a method that can dynamically update the ranking of answers in response to incremental feedback of preferences, while maintaining the structure of the initial scores provided by the base QA model. Following our running example, given a soft preference like "*awards involving performers*", we would like awards such as "*Best Actress*" to rank higher than "*Best Sound*", but both entities should still rank higher than incorrect entities, such as "*Nobel Prize*".

To achieve this, we propose to modify the original score of each entity with a linear function with three inputs: (1) the original score computed by the base QA model, (2) its average similarity to *positively* labeled entities, and (3) its average similarity with *negatively* labeled entities. Formally, we start by defining a partition of a preference set into the following two sets:

$$P^+ = \{e \mid (e, l) \in P(t), \, l = 1\}, \quad P^- = \{e \mid (e, l) \in P(t), \, l = 0\}, \tag{3}$$

where we have dropped the dependency on $t$ for brevity. Let $a[e]$ denote the score of entity $e$ extracted from the vector of scores $a(q)$ computed by the base QA model. Our proposed score update is given by the following expression:

$$a^{\text{new}}[e] = f(a[e], \delta(P^+, e), \delta(P^-, e)), \tag{4}$$

where $f$ is a linear function of its arguments and $\delta$ is a function that computes the mean cosine similarity of a preference set and an entity $e$:

$$\delta(P^\odot, e) = \frac{1}{|P^\odot|} \sum_{e_i \in P^\odot} \text{sim}(e_i, e). \tag{5}$$

The majority of existing QA models learn entity embeddings for computing query scores (Hamilton et al. (2018); Daza & Cochez (2020); Ren et al. (2020); Ren & Leskovec (2020); Arakelyan et al. (2021; 2023); Bai et al. (2023); Zhu et al. (2022), among others). We propose to reuse the embeddings of the base QA model (which are already required to compute the initial vector of scores $a(q)$) to compute the cosine similarity $\text{sim}(e_i, e)$ required in Equation (5).

Our proposed score update has two benefits: first, the use of a linear function **preserves monotonicity** in the scores, such that if two entities have equal similarities with respect to a preference set, then differences in their scores will be based exclusively on the initial scores computed by the QA model. Second, it is **efficient**, since it relies on already trained embeddings from the base QA model, and its application has linear time complexity, as we show next.

**Time complexity.** Let $n = |\mathcal{V}|$ be the number of entities, $d$ the embedding dimension, and $t = |P(t)|$ the number of preferences. For each entity $e \in \mathcal{V}$ we compute $\delta(P^\odot, e)$ as the mean of $t$ cosine similarities. Each cosine reduces to a dot product between $d$-dimensional vectors, so it costs $\mathcal{O}(d)$; hence one entity requires $\mathcal{O}(td)$ time to accumulate $t$ dot products and take their mean, and processing all $n$ entities costs $\mathcal{O}(ntd)$ time.

We now introduce two implementations of the update function $f$ in Equation (4): a linear combination with fixed coefficients, and a neural reranker where the coefficients are computed conditionally.

### 5.1 Linear with fixed coefficients (Cosine).

Motivated by prior work on reranking methods for search over knowledge graphs (Gerritse et al., 2020; Daza et al., 2021), we first propose a score update that is computed as a convex combination of the original score, and a linear combination of the average cosine similarities with $P^+$ and $P^-$. We refer to this as the **Cosine** update:

$$f_{\text{Cos}}(a[e], \delta(P^+, e), \delta(P^-, e)) = \alpha \cdot a[e] + (1 - \alpha)\left(\frac{1+\beta}{2}\delta(P^+, e) - \frac{1-\beta}{2}\delta(P^-, e)\right) \quad (6)$$

The cosine update thus requires tuning two hyperparameters, $\alpha \in (0, 1)$ and $\beta \in (-1, 1)$ that control the weight given to the original score, and the average cosine similarities, which can be done using a validation set.

### 5.2 Neural query reranker (NQR).

We can further adapt the weights given to each term in the score update function in Equation (6) by computing them with a small multi-layer perceptron (MLP) with four inputs: the original score $a[e]$, the average similarities $\delta(P^+, e)$ and $\delta(P^-, e)$, and the embedding $\mathbf{e} \in \mathbb{R}^d$ of entity $e$ (which we reuse from the base QA model and do not fine-tune).

Intuitively, making the weights conditional allows to contextualize the score update to specific values of scores, similarities, and the entity as captured by its embedding. The MLP outputs two scalars $\alpha_p, \alpha_n \in (0, 1)$, which we then use in the update rule (see Figure 2). We refer to this as the Neural Query Reranker (**NQR**) update:

$$f_{\text{NQR}}(a[e], \delta(P^+, e), \delta(P^-, e)) = a[e] + \alpha_p\delta(P^+, e) - \alpha_n\delta(P^-, e). \quad (7)$$

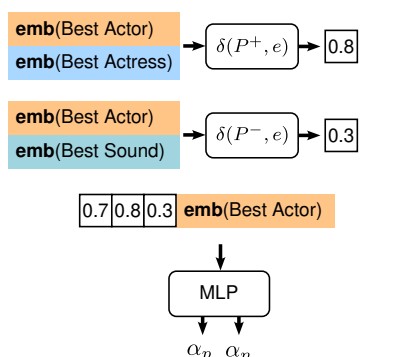

Figure 2: To adjust the score of an entity $e \in \mathcal{V}$ (*Best Actor*), NQR computes similarity scores with entities in the sets $P^+$ (*Best Actress*) and $P^-$ (*Best Sound*) using their embeddings (shown as emb). These scores, together with the original score (here, 0.7) and the embedding of $e$ are the input to an MLP that computes coefficients $\alpha_p$ and $\alpha_n$ for Equation (7).

**Time complexity.** If the hidden dimension of the MLP is $h$, the forward pass per entity multiplies a $(d + 3) \times h$ matrix and an $h \times 2$ matrix (plus activations), which is $\mathcal{O}(dh)$ time; across all entities this adds $\mathcal{O}(ndh)$. Thus the NQR update over all entities runs in $\mathcal{O}(n(td + dh))$ time, remaining linear in the number of entities in the KG.

**Training NQR.** The main goal of NQR is to adjust entity scores such that the information conveyed by the preference data is reflected in the ranking, while preserving the overall quality of the original query answers. This can be achieved with a training set of logical queries and their corresponding answers, together with a partition of the set of answers into disjoint preference sets $P^+$ and $P^-$. The training objective is then two-fold: (1) entities in $P^+$ should be ranked higher than those in $P^-$, and (2) all entities in $P^+ \cup P^-$ (which satisfy the original logical query) should still rank above entities that do not meet the logical constraints.

To achieve this, we adopt the RankNet loss (Burges et al., 2005), which encourages pairs of relevant and non-relevant items to be ordered correctly. Given two entities $e_i$ and $e_j$ with scores $a[e_i]$ and $a[e_j]$, RankNet minimizes the binary cross-entropy over the probability $\sigma(a[e_i] - a[e_j])$ that $e_i$ should be ranked above $e_j$. For NQR, we define two complementary loss terms:

$$\mathcal{L}_{\text{NQR}} = \underbrace{\mathbb{E}_{(e^+,e^-)\sim(P^+,P^-)}\big[-\log\sigma\big(a[e^+] - a[e^-]\big)\big]}_{\text{(1) Preference-based ranking}} + \underbrace{\mathbb{E}_{(e^+,e_r)\sim(P^+\cup P^-,\mathcal{N}_m)}\big[-\log\sigma\big(a[e^+] - a[e_r]\big)\big]}_{\text{(2) Query-consistency ranking}}, \quad (8)$$

where $\mathcal{N}_m$ is a set of $m$ randomly sampled negative entities from $\mathcal{V}$ that are not valid answers to the query. The first term enforces the ordering implied by the preference set, while the second term ensures that preferred and non-preferred but valid answers remain ranked higher than irrelevant entities. We use gradient descent on this loss to update the parameters of the MLP in NQR.

## 6 Experiments

We aim to evaluate to what extent we can incorporate soft constraints as expressed in preference sets, while preserving overall query answering performance. We extend current benchmarks for query answering with preference data, and we run experiments comparing the performance of the Cosine and NQR updates, together with a baseline based on gradient-boosting approaches for learning to rank.

### 6.1 Datasets

Prior work on query answering on KGs has relied on datasets containing query-answer pairs $(q, \mathcal{A})$, where $q$ is a query and $\mathcal{A} \subset \mathcal{V}$ is the set of answers computed over a KG (Ren et al., 2020; Ren & Leskovec, 2020). We extend these datasets by deriving preference data from observable clusters of entities in the set $\mathcal{A}$. Motivated by prior work showing that textual embeddings correlate with human notions of similarity and relatedness (Reimers & Gurevych, 2019; Grand et al., 2022; Merkx et al., 2022), we achieve this by clustering embeddings of answers to a query, which we use as a computational proxy of similarity that allows us to study the problem at a scale of thousands of queries and comprising different notions of similarity. Concretely, this process consists of three steps (we provide additional details in Appendix A):

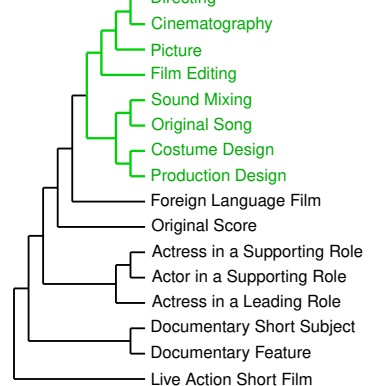

Figure 3: Example dendrogram used to generate preference sets. Highlighted in green are preferred entities.

**1. Embedding textual descriptions.** In order to capture soft similarity relationships that go beyond the structure of the graph and are harder to encode via logical constraints, we retrieve textual descriptions of entities in the KG and compute embeddings for each of them using a text embedding model. This results in clusters capturing information not available to the QA model or any of the reranking methods, which rely only on graph embeddings and have no access to textual descriptions.

**2. Clustering.** We apply Hierarchical Agglomerative Clustering to the entities in $\mathcal{A}$, using cosine similarity as the distance metric. As shown in Fig. 3, the result is a dendrogram that iteratively merges the most similar entities into clusters, capturing similarities at different levels.

**3. Answer set partitioning.** We traverse the dendrogram from the root node towards the leaves, and if a cluster contains at least 20% of the entities in the answer set[1], we use it to form a preference set. For each

---
[1]We define this limit to avoid clusters with too few entities.

Table 1: Statistics of the datasets used in our experiments.

| Dataset | Knowledge Graph | | | Queries | | | Preference Sets | | |
|---|---|---|---|---|---|---|---|---|---|
| | Entities | Relations | Edges | Train | Validation | Test | Train | Validation | Test |
| FB15k237 | 14,505 | 237 | 310,079 | 8,472 | 24,435 | 24,505 | 42,360 | 122,175 | 122,525 |
| Hetionet | 45,158 | 24 | 2,250,198 | 55,772 | 24,427 | 24,494 | 278,860 | 122,135 | 122,470 |

entity $e_i \in \mathcal{A}$, we define its preference label as $l_i = 1$ if it is in the cluster, and $l_i = 0$ otherwise. An example of a preference set is shown in Fig. 3, with entities labeled as $l_i = 1$ highlighted in green, and the rest of the entities labeled as $l_i = 0$. This partition forms the preference set $P(T) = \{(e_1, l_1), \ldots, (e_T, l_T)\}$. We then denote an instance in our extended dataset as $(q, \mathcal{A}, P(T))$, with $10 \leq T \leq 100$.

Prior work has relied on subsets of encyclopedic KGs such as NELL (Carlson et al., 2010) and Freebase (Bollacker et al., 2008) to benchmark methods for query answering. These KGs contain statements about people, organizations, and locations around the world. For our experiments, we select FB15k237 (Toutanova & Chen, 2015) and the complex queries generated for it by Ren & Leskovec (2020). To further explore the potential of query answering in a different domain we use Hetionet (Himmelstein et al., 2017), a biomedical KG containing ∼2M edges between genes, diseases, and drugs, among other types. For Hetionet, we extract complex queries of different types, and then for both datasets we run the three steps above to collect preference data. Furthermore, in order to validate the alignment between preference sets derived from embeddings and human notions of similarity, we manually generate clusters of answers to queries for a subset of 140 queries in the FB15k237 dataset, and then measure the accuracy of clusters from embeddings with respect to the manually created ones, which results in an agreement of 79% (see Appendix A for details).

We consider 14 types of first order logic queries of varying complexity, including negation. We present statistics of the final datasets and splits in Table 1. For training NQR, we only use 1-hop, link prediction queries. For the validation and test sets we use the full set of 14 types of complex queries. More detailed statistics can be found in Appendix B.

## 6.2 Evaluation

We evaluate performance on the task of interactive query answering with soft constraints as follows: given an instance $(q, \mathcal{A}, P(T))$ in the test set and a list of initial scores $a(q)$ computed by a base QA model, we select subsets of $P(t) \subseteq P(T)$ **incrementally**, with $1 \leq t \leq 10$. For each subset we compute the adjusted score (Eq. 4), and then we evaluate the following:

**1. Pairwise accuracy:** Measures the extent to which preferred entities are ranked higher than non-preferred entities. Let $\mathsf{rank}(e, a(q))$ be the position of entity $e$ when sorting the list of scores $a(q)$ in descending order. We define the pairwise accuracy as follows:

$$\text{PA} = \sum_{e^+ \in P^+} \sum_{e^- \in P^-} \mathbb{1}\left[\mathsf{rank}(e^+, a(q)) < \mathsf{rank}(e^-, a(q))\right], \tag{9}$$

where $\mathbb{1}[\cdot]$ is an indicator function. Importantly, while score updates are computed using a subset of $P(T)$ of maximum 10 preferences, PA is computed using the full set, which can contain up to 100 preferences. This allows us to evaluate whether methods can generalize from a limited number of examples.

**2. Ranking metrics:** Following prior work on CQA, we evaluate evaluate query answering performance using ranking metrics. For an answer $e \in \mathcal{A}$ to a query, the reciprocal rank is $1/\mathsf{rank}(e, a^{(t)})$, and the Mean Reciprocal Rank (MRR) is the average over all queries[2]. Hits at $k$ (H@k) is defined by verifying if, for each answer $e$, its rank $\mathsf{rank}(e, a^{(t)})$ is less than or equal to $k$ (in which case H@k = 1, and H@k = 0 otherwise), and averaging over all queries. We compute these ranking metrics using the original and adjusted scores to determine any effect on overall query answering performance.

---

[2]We compute *filtered* ranking metrics, where other known answers to a query are removed before computing $\mathsf{rank}(e, a^{(t)})$.

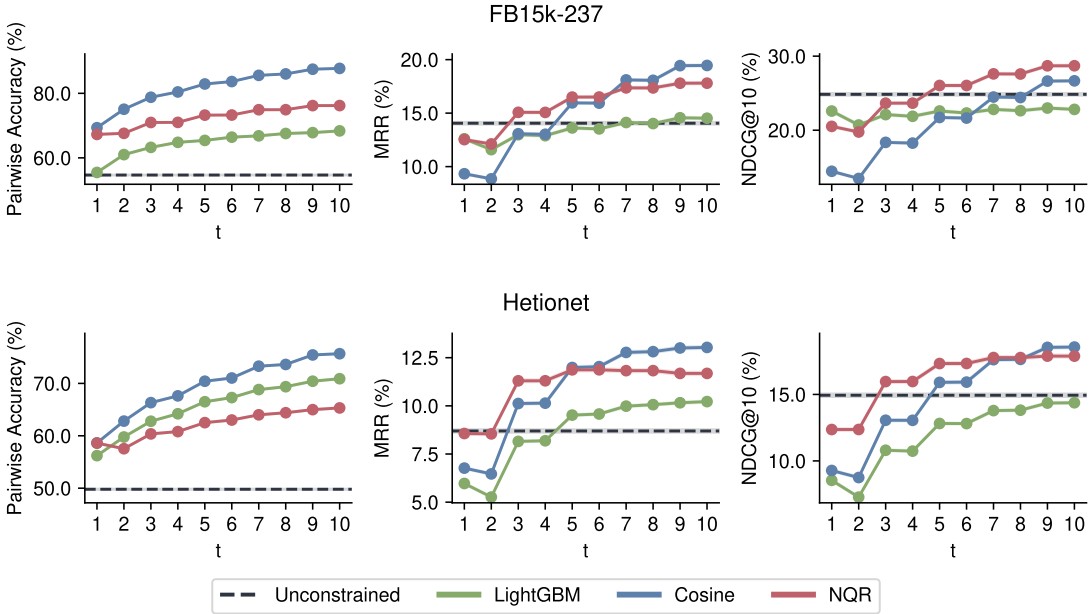

Figure 4: Results on interactive query answering with soft preferences, on FB15k237 and Hetionet.

**3. NDCG:** We further compute the Normalized Discounted Cumulative Gain at $k$ (NDCG@k), which jointly captures the two criteria measured by PA and traditional ranking metrics: the relative ordering of preferred versus non-preferred answers, and their positions within the ranked list. We assign relevance scores of $0, 1, 2$ to non-answers, answers in $P^-$, and in $P^+$, respectively. The discounted cumulative gain and its normalized version are

$$\text{DCG@}k = \sum_{e \in A_i(q)} \frac{2^{\text{rel}(e)} - 1}{\log_2(j+1)}, \qquad \text{NDCG@}k = \frac{\text{DCG@}k}{\text{IDCG@}k}, \qquad (10)$$

where IDCG@$k$ is the DCG of the ideal ranking. An NDCG@k of 1 indicates that all preferred answers appear above non-preferred and non-answers.

## 6.3 Baselines

We consider a traditional Learning-to-Rank (LTR) baseline based on LightGBM (Ke et al., 2017), a gradient boosting framework implementing the LambdaRank objective (Burges et al., 2005; 2006). For each training query, the model receives as input three scalar features for every entity $e$: the original score $a[e]$, and the average positive and negative similarities, $\delta(P^+, e)$ and $\delta(P^-, e)$. Graded relevance labels are assigned following standard LTR practice (Burges, 2010): entities in $P^+$ receive a label of 2, entities in $P^-$ receive 1, and a random set of 100 negative entities is assigned 0. The model is trained to predict ranking scores that order entities according to their relative relevance by directly optimizing NDCG via the LambdaRank objective. This baseline allows us to compare our proposed linear score update based on shifts from the original score, with an approach that predicts the score using a nonlinear function of relevant features. For all models, we find the best values of hyperparameters using the validation set via grid search. In all cases, we use QTO (Bai et al., 2023) as the base QA model. More details can be found in Appendix C.

## 7 Results

**Soft constraints and ranking quality.** We present results of pairwise accuracy, MRR, and H@10, and NDCG@10 in Fig. 4. For reference, we denote the metrics of the base QA model as **Unconstrained**, which allows us to determine how the adjustments in scores introduced by the methods we consider affect the

performance of the original scores. We include 95% confidence intervals as shaded regions around the curves; these are narrow and often indistinguishable, indicating that the differences are consistent across queries.

**Pairwise accuracy.** Across both datasets, we observe that all methods manage to achieve a pairwise accuracy significantly higher than the starting value (when no preferences are taken into account), which increases consistently as larger preference sets are provided. This indicates that all methods correctly adjust the scores in a way that entities in $P^+$ are ranked higher than those in $P^-$. Notably, the Cosine method achieves the highest values, which reaches a 87.7% in FB15k237, and 75.7% in Hetionet.

**Ranking metrics.** Ranking metrics (MRR and H@10) behave differently in comparison with PA. When preference sets are small, between 1 and 4 in size, score adjustments often cause a degradation in ranking quality with respect to the original scores. The fact that PA is already high at such sizes of preference sets indicates that the adjusted scores greedily place entities in $P^+$ above $P^-$, possibly by placing wrong answers between them, which harms ranking quality. Considering all methods, and especially in Hetionet, NQR causes the smallest drops in performance at small preference sets. As preference sets grow larger, the performance of all methods (with the exception of LightGBM, which struggles on FB15k237 and in H@10 on Hetionet) increases above the original. Both Cosine and NQR yield the highest ranking metrics, indicating that linear score updates are better for preserving ranking quality in comparison with the LightGBM baseline that predicts the new score directly.

**NDCG.** The NDCG@10 results provide a unified view of the two underlying criteria: pairwise ordering and overall ranking quality. For small preference sets, all constrained models achieve higher NDCG@10 than the Unconstrained ranking, primarily due to better pairwise accuracy. Since both PA and ranking metrics improve as the size of the preference sets increases, NDCG@10 exhibits the same upward trend. Importantly, NDCG@10 allows comparing which method best satisfies the dual objective of query answering under soft constraints. For small preference sets, NQR performs best, suggesting that its learned conditional weights help preserve ranking quality while applying fine-grained score adjustments. For larger sets, the Cosine update surpasses NQR, matching or improving ranking quality while achieving higher PA.

We present additional results in Appendix D where metrics are categorized according to query type. On average, Cosine performs best at balancing similarity constraints with global ranking performance, followed by NQR, which for certain query types, preserves best global ranking performance. Furthermore, we present results on manually curated preference sets, confirming the trends we observe when evaluating models on preference sets derived from embeddings, supporting their use to estimate human notions of similarity.

**Discussion.** Task-specific training of NQR is particularly beneficial for preference sets of size 4 or smaller, as it prevents large drops in ranking performance (MRR) while accurately capturing preferences (pairwise accuracy). This is especially relevant in settings where preference sets are costly to obtain, such as human annotation. Moreover, higher NDCG@10 for preference sets up to size 8 across both datasets suggests that NQR achieves a better balance between ranking quality and preference alignment.

These results highlight a key challenge in incorporating soft constraints into complex queries. Pairwise accuracy reflects whether preferred entities are ranked above non-preferred ones, whereas ranking metrics depend on the absolute positions of correct answers. With few preferences, the reranking signal is sufficient to separate preferred from non-preferred entities locally, improving pairwise accuracy, but not strong enough to consistently reorder the full ranking. This can temporarily promote non-answer entities, leading to lower MRR despite improved pairwise ordering. As more preferences are incorporated, the similarity signal becomes more stable and better aligned with the structure of the answer set. This reduces such spurious promotions and allows the ranking to recover, ultimately improving both preference alignment and global ranking metrics.

**Trade-offs induced by the Cosine update.** Figure 5 illustrates how pairwise accuracy (PA) and mean reciprocal rank (MRR) evolve jointly for different values of $\alpha$ and $\beta$ in the Cosine update (Equation (6)), as the number of interaction steps increases. Across both datasets, increasing $\alpha$ (which implies placing more weight on the original scores), prevents significant drops in MRR, but also limits the maximum PA that can be reached. Smaller $\alpha$ values, on the other hand, allow stronger preference propagation, and the effect

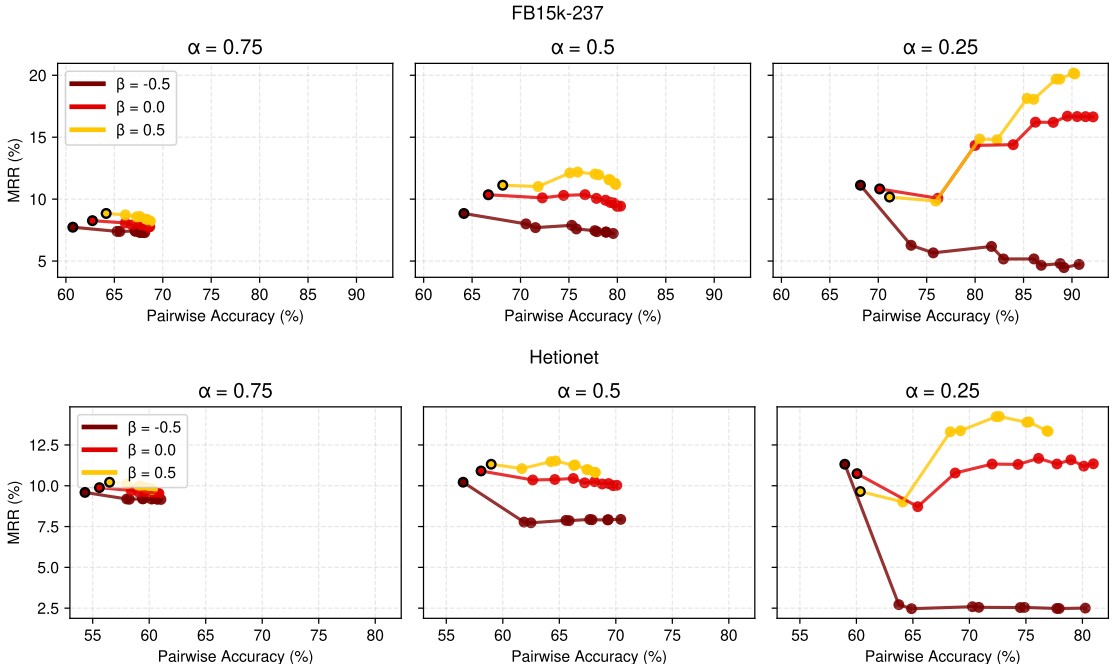

Figure 5: Trajectories of pairwise accuracy vs MRR for the Cosine update. Each subplot corresponds to a different value of $\alpha$, with lines colored by values of $\beta$. Highlighted circles indicate the initial interaction step (t=1), and the following points show increasingly larger preference sets.

on MRR varies depending on the value of $\beta$. This behavior reflects an important trade-off: higher $\alpha$ yields conservative and stable adjustments, while lower $\alpha$ enables more aggressive shifts guided by similarity signals.

**Actionable control via $\beta$.** Within each $\alpha$ setting, varying $\beta$ determines the relative emphasis placed on positive versus negative preferences. Larger $\beta$ values (yellow curves) consistently increase PA with limited loss in MRR, while negative $\beta$ values (dark red) degrade both, indicating that over-penalizing negative examples distorts the global ranking. When $\alpha = 0.25$ and after 10 interactions, the values closer to the Pareto front are $\beta = 0$ and $\beta = 0.5$. Values of $\beta = 0.5$ increase PA further, at the cost of lower MRR, and viceversa for $\beta = 0$. This shows that $\beta$ offers an interpretable control mechanism: users prioritizing ranking fidelity can select lower $\beta$ values to protect MRR, whereas larger $\beta$ values lead to stronger preference alignment by boosting PA. This tunability makes the Cosine update useful for downstream applications, enabling experts to adjust the balance between preserving original rankings and enforcing similarity constraints according to their needs.

**Qualitative example.** Table 3 shows a 2-hop query from FB15k237, where the task is to rank awards associated with people born in New York City. Applying similarity constraints on the target variable using the Cosine update sharply improves ranking quality: literary awards in the positive set are promoted into the Top-5, while unrelated film and TV awards drop in rank. NDCG@10 increases from 0.0 to 0.60, illustrating how a small preference set can adjust the ranking towards the soft constraint that awards returned as an answer to the query should be literary rather than film or music awards. We also observe a failure mode in which the process of reranking can cause correct answers to be demoted below incorrect entities. We show the affected entities, together with their drop in position from the original ranking. This highlights the trade-off that arises when incorporating soft constraints: reranking improves alignment with the soft constraint, but can also over-propagate similarity signals and harm some correct answers, especially with limited preferences.

**Runtime.** We present the average runtime in milliseconds per query in the FB215k237 dataset for all methods in Table 2, where we include the added runtime ($\Delta$) with respect to the Unconstrained baseline. We note that LightGBM results in very high overhead, by adding 13.8 ms per query, due to the use of sequential tree traversal.

Table 3: Example 2-hop query from FB15k237, representing the question *"Which awards have had nominees who were born in New York City?"* and specific preference sets for it. We show the top-5 ranked target entities before and after applying similarity constraints using the Cosine update, and answers incorrectly demoted by this process together with their drop in position in the ranking (shown in parenthesis). ▲ indicates a promoted entity and ▼ a demoted one. $P^+$ and $P^-$ denote preference sets; $v_1, v_2$ variables.

| $q(v_1, v_2) = $ PlaceOfBirth(New York City, $v_1$) $\land$ NominatedFor($v_1, v_2$) | | |
|---|---|---|
| $P^+$ | | $P^-$ |
| Hugo Award for Best Novel | | Grammy Award for Best Rock Instrumental Performance |
| World Fantasy Award for Best Novel | | Grammy Award for Best Country Song |
| World Fantasy Award for Best Novella | | Grammy Award for Best Country Performance |
| **Initial Top-5** | **Top-5 after update** | **Incorrectly demoted answers** |
| Original Score, Golden Globe | ▲ Novel, Nebula | ▼ Individual Performance, Emmy (-15) |
| Original Song, Academy | ▲ Novella, Hugo | ▼ Worst New Star, Razzie (-4) |
| Best Director, Academy | ▲ Short Story, Hugo | ▼ Outstanding Supporting Actress, Emmy (-22) |
| Original Screenplay, Academy | ▲ Professional Artist, Hugo | ▼ Academy Award for Best Original Musical (-13) |
| Nobel Prize in Literature | ▲ Novel, Hugo | ▼ Original Screenplay, Writers Guild (-16) |
| NDCG@10: 0.0 | NDCG@10: 59.9 | |

On the other hand, the Cosine and NQR methods add very little overhead of up to 1.4 ms per query. While the average runtime of Cosine seems higher than NQR, a paired t-test does not show that the difference is statistically significant.

**Reranking with large language models.** Finally, we consider the use of large language models (LLMs) when given access to textual descriptions (which we do not assume in our reranking experiments). Importantly, LLMs are limited by the high computational cost required to encode text where memory consumption increases quadratically (Vaswani et al., 2017). To quantify this, we measure the runtime of an LLM when cluster-

Table 2: Average execution time per query for different reranking methods and difference ($\Delta$) with Unconstrained.

| Method | Runtime (ms) | $\Delta$ (ms) |
|---|---|---|
| Unconstrained | 0.6 | — |
| LightGBM | 14.4 | +13.8 |
| Cosine | 2.0 | +1.4 |
| NQR | 1.8 | +1.2 |

ing the set of answers to a query (instead of reranking *all* entities in the KG, as our experiments require), and we find that the average runtime per answer set is 3.5 seconds in average over 140 queries (95% CI [3.3, 3.8]). While this includes network effects, it is much larger than the slower method in our experiments which require reranking betweeen 14k to 45k entities per query. This motivates future work on applying LLMs for incorporating soft constraints.

## 8 Conclusion

We have introduced the problem of interactive knowledge graph query answering with soft entity constraints. We formulate it as an extension that addresses the limitations of logical queries over knowledge graphs, which are restricted by the expressivity of first order logic and the schema of the graph. We introduce efficient methods for incorporating soft constraints in complex queries. Since they reuse embeddings of a base CQA model, the methods are lightweight, requiring tuning only two parameters or a small MLP. To evaluate these models over a large and diverse set of queries and notions of similarity, we extend existing query answering benchmarks with preference sets derived from clusterings of embeddings. In our experiments, we find that these methods are able to adjust query scores to capture preferences about their answers, while increasing ranking performance as more exemplary preferences are provided. In practice, we observe that our methods are fast, adding only up to 1.4 ms overhead to the base QA model. With soft constraints, we explore a new and flexible way to interact with graph databases, which can allow users to specify their preferences by providing examples interactively, leading to query answering systems that can quickly adapt to user feedback.

## Acknowledgment

Daniel Daza was funded by a gift from Accenture LLP. Michael Cochez is partially funded by the Elsevier Discovery Lab. His work on this publication is in part based upon work from COST Action CA23147 GOBLIN - Global Network on Large-Scale, Cross-domain and Multilingual Open Knowledge Graphs, and COST Action CA24121 - Knowledge Graphs in the Era of Large Language Models (KGELL), both supported by COST (European Cooperation in Science and Technology, https://www.cost.eu).

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

## Appendix

## A    Generating preference data

In our experiments, we obtain preference data by clustering answers to queries over a KG using embedding models that map the description of an entity to an embedding of fixed dimension. In this section we provide additional technical details of this process.

**Embedding textual descriptions.**    We rely on text embedding models publicly available on Hugging Face[3] for computing embeddings of entities based on their textual descriptions. For entities in FB15k237, we use `NovaSearch/stella_en_400M_v5`[4], with an embedding dimension of 1,024. For Hetionet we employ `Alibaba-NLP/gte-Qwen2-7B-instruct`[5], which at the time of writing is ranking 1st in the Massive Text Embedding Benchmark[6] in the medical domain. The embedding dimension of this model is 3,584.

**Clustering.**    We execute hierarchical clustering using the average linkage method and cosine distance as the dissimilarity metric, applied to the target embeddings. This is implemented using the `linkage` function from the SciPy package. Average linkage is selected because it considers the mean pairwise distance between points in different clusters, which tends to produce more balanced and interpretable cluster structures. Cosine distance is used as it effectively captures angular similarity, which is more meaningful than Euclidean distance in high-dimensional embedding spaces, where the direction of vectors often encodes more relevant information than their magnitude.

**Validation.**    The use of embedding models as proxies of similarity is supported by prior works indicating their correlation with human judgments (Reimers & Gurevych, 2019; Grand et al., 2022; Merkx et al., 2022). In order to quantify this correlation in the context of preference sets generated via embeddings, we build a set of manually curated preference sets for queries in the FB15k237 dataset, and we then measure how well they overlap with our preference sets derived from embeddings. This process consists of three steps:

1. **Candidate generation.** Generating partitions of query answer sets can be a very challenging task, as a single query can have tens to hundreds of answers and several different ways to define a partition, resulting in very long annotation times. To make this study feasible, we rely on a large language model (LLM) to propose an initial candidate partition, which is then approved by a human. In particular, for a given query and its set of answers, we prompt an LLM (GPT 5.1) to propose a partition of the answer set into positives and negatives by relying on a vague property that is hard to formalize. The full prompt is as follows:

   > "You are clustering entities by a human-interpretable shared property that is difficult to formalize. Given candidate entity descriptions, choose one coherent positive group and assign the rest to negatives. Prefer positives that share a specific latent theme, not a generic category that can be easily found in a knowledge graph. Only rely on the explicit information provided about each entity, not on background knowledge. Return only valid JSON with keys: label, rationale, positives, negatives. The positives and negatives must be lists of integer candidate indices. Positives must be non-empty, negatives must be non-empty, and together they must cover each candidate exactly once."

2. **Manual selection.** A human (co-author in this study) is asked to confirm whether each candidate is a valid partition of the answer set. The annotation guideline is the following: *"a clustering is valid if the entities in the positive set share a common property not represented by the entities in the negative set, and that is difficult to formalize with first-order logic."* We repeat this process until we collect preference sets for 10 queries of each of the 14 types we consider in our work (see Figure 6), for a total of 140 queries. We present examples of the rationales for the selected preference sets in Table 4.

---

[3]https://huggingface.co/
[4]https://huggingface.co/NovaSearch/stella_en_400M_v5
[5]https://huggingface.co/Alibaba-NLP/gte-Qwen2-7B-instruct
[6]https://huggingface.co/spaces/mteb/leaderboard

Mid-sized interior Western US cities (not state capitals) near mountains or high plains
Primarily known as record producers / mixing engineers rather than as front-facing recording artists
Films adapted from literary works that are *not* straightforward biographies or autobiographies
Primarily visual artists rather than musicians/performers
American screenwriters/playwrights primarily known for writing (not acting) for film/TV or stage
Primarily celebrated as innovative lead guitarists in major rock bands rather than as solo singer-songwriters
Biographical dramas about real historical/political figures (film or TV)
American film/TV composers (primarily known for composing scores rather than performing)
Rock subgenres defined by blending rock with electronic/technological elements or dance music
Writers strongly associated with speculative genres (fantasy, science fiction, or horror)

Table 4: Examples of human-validated rationales for preference sets.

| Query Type | Accuracy | 95% CI |
|---|---|---|
| 1p | 0.8560 | [0.7687, 0.9410] |
| 2p | 0.8498 | [0.7663, 0.9234] |
| 3p | 0.8238 | [0.7448, 0.9009] |
| 2i | 0.8663 | [0.7963, 0.9361] |
| 3i | 0.7175 | [0.6197, 0.8068] |
| ip | 0.8778 | [0.8097, 0.9436] |
| pi | 0.8376 | [0.7638, 0.9038] |
| 2in | 0.7640 | [0.7003, 0.8344] |
| 3in | 0.7588 | [0.6825, 0.8459] |
| inp | 0.7807 | [0.7250, 0.8364] |
| pin | 0.7818 | [0.7092, 0.8588] |
| pni | 0.6611 | [0.6328, 0.6874] |
| 2u-DNF | 0.6539 | [0.5996, 0.7104] |
| up-DNF | 0.8364 | [0.7683, 0.8980] |

Table 5: Accuracy computed between the curated and embedding-based preference sets, broken down by query type.

3. **Agreement evaluation.** For each of the 140 queries, we compare the embedding-based partition of its answer set with the manually curated partition. Treating the manual annotation as ground truth, we compute accuracy as the fraction of entities whose positive/negative assignment matches between the two partitions. Across all 140 annotated queries, the mean accuracy between the curated preference sets and the embedding-based ones is **0.7904**, with 95% CI [**0.7677, 0.8129**]. Table 5 shows the breakdown by query type. Agreement is strongest for path/intersection families such as `ip` (0.8778), `2i` (0.8663), `1p` (0.8560), and `2p` (0.8498), while union and negation-heavy families remain more challenging, notably `2u-DNF` (0.6539) and `pni` (0.6611).

The observed accuracy thus indicates a substantial match between preference sets derived from embeddings and from human notions of similarity, supporting the use of synthetic clusters as a practical proxy for training and evaluating models at the scale of thousands of queries, where fully manual annotation would be prohibitively expensive.

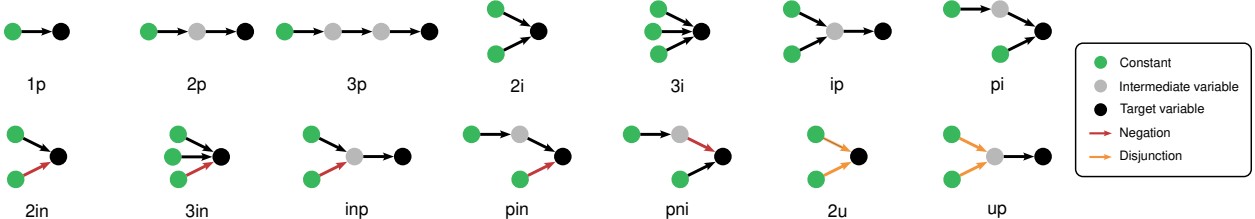

Figure 6: Types of complex queries used in our experiments.

Table 6: Statistics of the datasets for query answering with soft preferences used in our experiments.

| Split | Structure | 1p | 2p | 3p | 2i | 3i | ip | pi | 2in | 3in | inp | pin | pni | 2u | up | Total |
|---|---|---|---|---|---|---|---|---|---|---|---|---|---|---|---|---|
| | | | | | | | FB15k237 | | | | | | | | | |
| Train | Queries | 8,472 | | | | | | | | | | | | | | 8,472 |
| | Preferences | 42,360 | | | | | | | | | | | | | | 42,360 |
| Valid | Queries | 281 | 2,120 | 2,660 | 779 | 410 | 2,054 | 1,211 | 1,793 | 1,439 | 2,230 | 2,523 | 2,074 | 2,095 | 2,766 | 24,435 |
| | Preferences | 1,405 | 10,600 | 13,300 | 3,895 | 2,050 | 10,270 | 6,055 | 8,965 | 7,195 | 11,150 | 12,615 | 10,370 | 10,475 | 13,830 | 122,175 |
| Test | Queries | 327 | 2,101 | 2,717 | 843 | 500 | 1,942 | 1,184 | 1,805 | 1,569 | 2,257 | 2,460 | 2,031 | 2,052 | 2,717 | 24,505 |
| | Preferences | 1,635 | 10,505 | 13,585 | 4,215 | 2,500 | 9,710 | 5,920 | 9,025 | 7,845 | 11,285 | 12,300 | 10,155 | 10,260 | 13,585 | 122,525 |
| | | | | | | | Hetionet | | | | | | | | | |
| Train | Queries | 55,772 | | | | | | | | | | | | | | 55,772 |
| | Preferences | 278,860 | | | | | | | | | | | | | | 278,860 |
| Valid | Queries | 1,469 | 2,319 | 2,573 | 802 | 491 | 2,116 | 1,365 | 1,564 | 956 | 2,475 | 2,557 | 1,142 | 1,936 | 2,662 | 24,427 |
| | Preferences | 7,345 | 11,595 | 12,865 | 4,010 | 2,455 | 10,580 | 6,825 | 7,820 | 4,780 | 12,375 | 12,785 | 5,710 | 9,680 | 13,310 | 122,135 |
| Test | Queries | 1,472 | 2,309 | 2,562 | 850 | 556 | 2,087 | 1,376 | 1,556 | 1,004 | 2,473 | 2,535 | 1,137 | 1,938 | 2,639 | 24,494 |
| | Preferences | 7,360 | 11,545 | 12,810 | 4,250 | 2,780 | 10,435 | 6,880 | 7,780 | 5,020 | 12,365 | 12,675 | 5,685 | 9,690 | 13,195 | 122,470 |

## B  Dataset statistics

We consider 14 types of complex queries over KGs introduced in prior work (Ren & Leskovec, 2020), which include conjunctions, disjunctions, and negation. These queries can be represented using query graphs, which we illustrate in Fig. 6.

In our experiments we consider complex queries over the FB15k237 and Hetionet KGs, selecting those containing at least 10 answers and at most 100. For each query, we execute the algorithm for generating preference data (Appendix A) and generate 5 preference sets per query. The final number of queries and preference sets for each KG is shown in Table 6.

## C  Experimental details

We implement all of our experiments using PyTorch (Paszke et al., 2019), and run them on a machine with one NVIDIA A100 GPU and 120GB of RAM.

**Base query answering model.**  We make use of QTO as the base query answering model Bai et al. (2023). QTO relies on a neural link predictor, for which we use ComplEx (Trouillon et al., 2016; Lacroix et al., 2018) trained with an auxiliary relation prediction objective (Chen et al., 2021). We use an embedding dimension of 1,000, learning rate of 0.1, batch size 1,000, and a regularization weight of 0.05.

**Cosine similarity.**  We tune the hyperparameters of this method using grid search, where $\alpha$ can take values in $\{0.25, 0.5, 0.75\}$, and $\beta$ in $\{-0.5, 0, 0.5\}$. We select the best values based on performance on the validation set. We define performance as the sum of pairwise accuracy and ranking performance.

**NQR.**  We describe the architecture of NQR in Table 7. We select the best learning rate in $\{1 \times 10^{-5}, 1 \times 10^{-4}, 1 \times 10^{-3}\}$.

Table 7: Layer-wise description of the model architecture in NQR. The embedding dimension is 1,000, and in the input layers we concatenate scalars (the current entity score, and the mean similarity scores with the positive and negative preference sets), indicated as +3.

|  | Input Dimension | Output Dimension |
|---|---|---|
| Linear + ReLU | 1000 + 3 | 256 |
| Linear + sigmoid | 256 | 2 |

Table 8: Per query type results of interactive query answering with feedback on the FB15k237 dataset.

| Method | 1p | 2p | 3p | 2i | 3i | ip | pi | 2in | 3in | inp | pin | pni | 2u | up | avg |
|---|---|---|---|---|---|---|---|---|---|---|---|---|---|---|---|
| Average Pairwise Accuracy | | | | | | | | | | | | | | | |
| Unconstrained | 58.44 | 55.67 | 55.30 | 55.55 | 53.45 | 56.19 | 54.97 | 58.06 | 54.99 | 53.41 | 54.40 | 58.70 | 59.08 | 55.74 | 56.00 |
| LightGBM | 80.10 | 66.07 | 59.85 | 71.98 | 66.97 | 61.55 | 61.70 | 74.45 | 69.43 | 61.25 | 62.89 | 73.14 | 75.09 | 67.58 | 68.00 |
| Cosine | **85.66** | **85.06** | **85.80** | **81.42** | **77.23** | **84.68** | **82.70** | **82.99** | **81.64** | **84.61** | **84.45** | **83.60** | **83.54** | **83.99** | **83.38** |
| NQR | 76.80 | 76.11 | 77.45 | 71.75 | 68.76 | 76.56 | 74.09 | 73.78 | 71.99 | 74.66 | 75.29 | 74.34 | 73.71 | 74.27 | 74.26 |
| Average MRR | | | | | | | | | | | | | | | |
| Unconstrained | 19.97 | 16.32 | 15.73 | 24.02 | 27.73 | **21.90** | **22.27** | 11.32 | 15.39 | 11.55 | 9.53 | 3.88 | 16.08 | 16.44 | 16.58 |
| LightGBM | 16.93 | 15.34 | 14.69 | 19.82 | 24.25 | 18.90 | 19.31 | 11.81 | 15.18 | 12.03 | 10.85 | **8.96** | 13.28 | 14.96 | 15.45 |
| Cosine | 24.09 | 16.65 | 14.52 | 23.25 | 27.78 | 16.77 | 17.51 | **15.08** | 16.31 | 14.47 | 13.02 | 7.89 | 17.88 | 16.12 | 17.24 |
| NQR | **25.37** | **19.08** | **16.28** | **26.22** | **29.17** | 19.15 | 20.34 | 14.60 | **17.79** | **15.98** | **13.11** | 5.63 | **18.37** | **18.84** | **18.57** |
| Average NDCG@10 | | | | | | | | | | | | | | | |
| Unconstrained | 39.92 | 36.79 | 36.94 | 51.40 | 54.67 | 44.44 | 48.28 | 33.56 | 43.10 | 29.86 | 28.28 | 10.86 | 43.37 | 39.88 | 38.67 |
| LightGBM | 45.62 | 43.73 | 42.06 | 53.42 | 56.09 | 48.13 | 51.49 | 43.07 | 49.81 | 39.32 | 38.45 | **36.44** | 45.42 | 46.72 | 45.70 |
| Cosine | **58.64** | 55.67 | 53.23 | **62.01** | **63.42** | 56.37 | 56.52 | **54.77** | **56.52** | 53.36 | **51.76** | 33.15 | **60.11** | 56.84 | **55.17** |
| NQR | 57.07 | **56.39** | **54.71** | 61.95 | 62.79 | **57.66** | **58.33** | 49.37 | 55.30 | 51.77 | 48.81 | 20.87 | 56.70 | **57.00** | 53.48 |

# D  Additional results

In our main experiments, we report the performance of different methods when presented with incremental feedback with preference sets of size 1 up to 10, averaged over all queries. This requires queries that have at preference set (combining both preferred and non-preferred entities) of at least 10 entities, such that we can evaluate methods over the entire range of preference set sizes.

We present here results over all types of complex queries we consider in our experiments. As in our main experiments, we compute the metrics for preference sets of increasing sizes, from 1 up to 10. Here, we report the metrics averaged over all 10 steps. We present results categorized by query structure (illustrated in Fig. 6) in Tables 8 and 9.

We observe that in average, Cosine and NQR result in the best results, with Cosine achieving a good balance between pairwise accuracy and global ranking performance, as shown by the results in NDCG@10. The results by query type also indicate cases where all models struggle to maintain the ranking performance of the Unconstrained model: in the FB15k237 dataset and for ip and pi queries, all models result in lower MRR, with NQR being the closest. Similarly, in Hetionet all models underperform the Unconstrained baseline in 2u queries, though NQR achieves a close value.

## D.1  Results on curated queries

We additionally evaluate the models on the subset of queries and manually curated preference sets obtained during the validation step described in Appendix A. The performance plots when increasing the preference set size are shown in Figure 7. Given the smaller sample size in this curated subset, we obtain wider confidence intervals. However, we observe that the general trends observed from our experiments using larger preference sets derived from embeddings hold.

We present more detailed results on the subset of curated queries in Table 10. When comparing these results with performance on preference sets derived from embeddings (see Table 8), we observe broadly consistent

Table 9: Per query type results of interactive query answering with feedback on the Hetionet dataset.

| Method | 1p | 2p | 3p | 2i | 3i | ip | pi | 2in | 3in | inp | pin | pni | 2u | up | avg |
|---|---|---|---|---|---|---|---|---|---|---|---|---|---|---|---|
| **Average Pairwise Accuracy** | | | | | | | | | | | | | | | |
| Unconstrained | 51.75 | 50.73 | 50.55 | 51.78 | 51.19 | 51.00 | 51.06 | 48.03 | 41.51 | 49.99 | 48.84 | 51.73 | 52.62 | 50.58 | 50.10 |
| LightGBM | **79.18** | 65.73 | 63.76 | **75.71** | **75.20** | 64.63 | 69.48 | 71.17 | 68.62 | 63.37 | 64.24 | 74.14 | **74.24** | 67.16 | 69.76 |
| Cosine | 75.27 | **73.30** | **71.65** | 72.13 | 71.48 | **73.19** | **73.57** | **74.19** | **74.58** | **73.29** | **71.12** | **75.06** | 70.70 | **69.21** | **72.77** |
| NQR | 68.17 | 63.19 | 60.78 | 65.96 | 65.30 | 65.40 | 67.12 | 66.84 | 69.39 | 61.60 | 60.67 | 67.28 | 63.44 | 59.59 | 64.62 |
| **Average MRR** | | | | | | | | | | | | | | | |
| Unconstrained | 25.91 | 4.83 | 7.86 | 21.74 | 26.20 | 7.98 | 15.00 | 13.27 | 8.53 | 4.25 | 3.80 | 3.27 | **18.89** | 5.69 | 11.95 |
| LightGBM | 13.37 | 8.47 | **11.31** | 11.42 | 14.28 | 10.26 | 11.47 | 9.26 | 7.87 | 8.19 | 7.23 | **9.87** | 9.19 | 7.32 | 9.97 |
| Cosine | 22.07 | **9.26** | 9.66 | 20.40 | 24.45 | 10.61 | 13.94 | 13.52 | 12.75 | **8.77** | **8.30** | 9.24 | 15.06 | **8.23** | 13.30 |
| NQR | **26.39** | 8.87 | 9.32 | **22.89** | **27.23** | **10.77** | **15.51** | **15.11** | **13.20** | 7.82 | 7.66 | 7.23 | 18.72 | 8.22 | **14.21** |
| **Average NDCG@10** | | | | | | | | | | | | | | | |
| Unconstrained | 44.20 | 19.83 | 26.34 | 43.06 | 45.51 | 24.72 | 38.31 | 33.33 | 22.95 | 19.29 | 17.58 | 9.62 | 48.75 | 23.92 | 29.81 |
| LightGBM | 41.45 | 41.17 | **49.29** | 39.12 | 41.06 | 43.87 | 40.70 | 35.96 | 29.84 | 43.06 | 39.60 | **38.24** | 40.35 | 43.64 | 40.52 |
| Cosine | **53.28** | **45.09** | 49.16 | **53.79** | **56.23** | **46.80** | 47.95 | **46.59** | **44.64** | **46.79** | **44.57** | 35.94 | 52.21 | **47.01** | **47.86** |
| NQR | 53.04 | 42.28 | 41.44 | 53.16 | 55.79 | 45.77 | **48.91** | 45.97 | 43.34 | 40.97 | 40.23 | 27.65 | **54.24** | 43.83 | 45.47 |

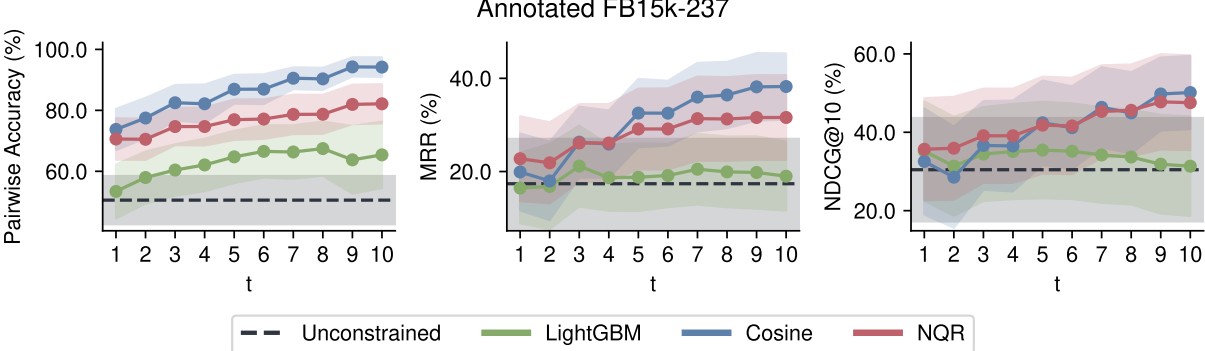

Figure 7: Results on interactive query answering with soft preferences on the subset of manually curated preference sets from the FB15k237 dataset.

trends in the relative performance of the methods, suggesting that results on the generated preference sets are reasonably predictive of behavior on the curated ones.

In particular, the Cosine baseline achieves the highest pairwise accuracy on both datasets, with a clear margin over the other methods, indicating that similarity-based reranking aligns well with both synthetic and human preferences. NQR consistently ranks second in pairwise accuracy and exhibits the strongest performance in ranking metrics: it achieves the best average MRR on both datasets and remains competitive in NDCG@10, often close to the Cosine baseline.

Importantly, the relative ordering of methods is largely preserved between datasets: LightGBM provides moderate performance, while Cosine and NQR dominate. At the same time, the curated dataset generally yields higher pairwise accuracy and ranking scores for the top-performing methods, suggesting that certain preference sets derived from embeddings may be harder to model due to imperfect clusters. Overall, these patterns indicate a meaningful correlation between the synthetic and curated evaluations, supporting the use of our setup for generating preference sets as a proxy for estimating real-world performance.

Table 10: Per query type results of interactive query answering with feedback on the subset of manually curated preference sets from the FB15k237 dataset.

| Method | 1p | 2p | 3p | 2i | 3i | ip | pi | 2in | 3in | inp | pin | pni | 2u | up | avg |
|---|---|---|---|---|---|---|---|---|---|---|---|---|---|---|---|
| | Average Pairwise Accuracy | | | | | | | | | | | | | | |
| Unconstrained | 39.66 | 54.92 | 53.16 | 42.82 | 55.02 | 58.34 | 55.71 | 58.23 | 39.35 | 52.46 | 40.32 | 34.87 | 42.43 | 44.96 | 48.02 |
| LightGBM | 73.41 | 73.46 | 66.20 | 61.90 | 73.47 | 55.11 | 62.31 | 83.61 | 66.28 | 53.00 | 63.46 | 63.01 | 58.37 | 66.54 | 65.72 |
| Cosine | **87.68** | **92.27** | **95.26** | **89.09** | **86.18** | **91.49** | **93.35** | **91.38** | **82.53** | **89.34** | **91.20** | **89.31** | **87.62** | **91.09** | **89.84** |
| NQR | 76.04 | 90.40 | 93.59 | 79.38 | 76.49 | 88.58 | 88.12 | 90.67 | 77.19 | 86.44 | 85.03 | 79.88 | 81.24 | 84.80 | 84.13 |
| | Average MRR | | | | | | | | | | | | | | |
| Unconstrained | 19.22 | 17.46 | 15.54 | 30.45 | 40.27 | 24.53 | 20.59 | 8.04 | 14.69 | 6.97 | 7.28 | 1.93 | 14.51 | 7.36 | 16.35 |
| LightGBM | 16.60 | 18.76 | 19.57 | 27.32 | 37.69 | 26.02 | 14.22 | 17.35 | 20.61 | 10.22 | 11.52 | 6.89 | 14.15 | 15.01 | 18.28 |
| Cosine | 22.90 | 18.93 | 23.00 | **37.61** | 38.16 | 26.90 | 21.77 | 19.09 | **22.03** | 19.92 | **17.28** | **10.49** | **22.08** | **22.38** | 23.04 |
| NQR | **27.54** | **23.24** | **23.57** | 37.50 | **42.10** | **27.17** | **25.02** | **20.26** | 20.55 | **20.03** | 15.89 | 4.53 | 20.67 | 19.37 | **23.39** |
| | Average NDCG@10 | | | | | | | | | | | | | | |
| Unconstrained | 32.99 | 28.45 | 27.83 | 38.17 | 58.53 | 37.63 | 35.48 | 18.97 | 27.06 | 16.39 | 17.51 | 4.05 | 24.15 | 12.58 | 27.13 |
| LightGBM | 43.87 | 39.68 | 40.80 | 47.51 | 65.06 | 43.67 | 37.18 | 44.33 | 46.48 | 25.32 | 29.57 | 24.32 | 33.94 | 37.47 | 39.94 |
| Cosine | **60.65** | 50.59 | **59.01** | **63.31** | **66.74** | **57.80** | 56.59 | 52.00 | **53.93** | **55.02** | **53.09** | **39.22** | **56.85** | **56.53** | **55.81** |
| NQR | 59.27 | **56.98** | 58.29 | 59.97 | 66.56 | 57.56 | **59.64** | **52.46** | 49.22 | 53.78 | 49.10 | 17.79 | 51.53 | 48.76 | 52.92 |

