# OpenReview forum: "Interactive Query Answering on Knowledge Graphs with Soft Entity Constraints"
_TMLR — Accepted by TMLR_

### Review · Reviewer_2YFJ · 2025-12-23

**Summary Of Contributions:**

This paper introduces the problem of "query answering with soft entity constraints" on knowledge graphs (KGs), aiming to supplement traditional first-order logic (FOL) queries with vague, user-specified preferences. The key contributions are:
1.  **Problem Formulation**: It defines and formalizes an interactive query answering setting where users provide a small set of positive and negative examples to represent a "soft constraint."
2.  **Lightweight Methods**: It proposes two efficient re-ranking methods, a linear combination (`Cosine`) and a small neural network (`NQR`), to adjust scores from a base QA model to reflect these soft constraints.
3.  **Benchmark Construction**: It presents a novel methodology to automatically generate benchmark datasets for this new task by applying hierarchical clustering on the text embeddings of known answers.

**Strengths**: The paper identifies a practical and valuable problem in making KG QA systems more interactive and personalized. The proposed methods are designed for efficiency, and the approach to benchmark creation is a pragmatic first step.

**Weaknesses**: The core technical novelty is limited, as the proposed methods are adaptations of standard re-ranking techniques. The formalism of key concepts like "soft constraint" is weak. The experimental validation relies entirely on a synthetic proxy for user preferences, which is a major limitation.

**Additional Comments:**

The problem of incorporating vague, example-based preferences into formal query answering systems on KGs is timely and important. The authors have done a good job of identifying this gap and proposing a straightforward evaluation framework. However, the current manuscript lacks the technical depth, novelty, and experimental rigor. The core weaknesses lie in the insufficient formalism of the problem, the limited novelty of the proposed re-ranking methods, and the reliance on an unvalidated synthetic dataset. I believe a major revision that addresses the critical requested changes is necessary before this work can be considered for publication.

**Audience:**

Yes

**Audience Explanation:**

Yes. The problem formulation itself is interesting and relevant to researchers in knowledge graph query answering, interactive machine learning, and personalized information retrieval. The idea of allowing users to refine complex logical queries with vague, example-based preferences is a practical and forward-looking direction. Even though the proposed solutions are not highly novel and the evaluation has limitations, the paper's articulation of this problem and its approach to creating a benchmark dataset could inspire further research in the TMLR community on more sophisticated methods for this valuable task.

**Broader Impact Concerns:**

The paper does not present significant broader impact concerns beyond those common to all personalization and re-ranking systems.

**Claims And Evidence:**

No

**Claims Explanation:**

The claims are not convincingly supported for several reasons, stemming from a lack of rigor in definitions, technical details, and experimental validation.

1.  **Weak Conceptual Foundation**: The central distinction between "hard" and "soft" constraints is not formally established. The paper claims soft constraints cannot be expressed in FOL, but fails to convincingly argue why a user's preference (e.g., "awards about audio-visual design") could not be modeled by extending the KG schema with a new predicate. The definitions are informal and rely on intuition rather than rigorous criteria.

2.  **Contradictory Claims about Objective**: The paper claims a goal is to "maintain the original ranking structure," but this is inherently at odds with the primary objective of re-ranking based on new preferences. The authors do not formally define what "preserving the structure" means, nor do they provide a metric to measure it, making the claim unsubstantiated.

3.  **Unsupported Experimental Premise**: The entire experimental validation rests on the assumption that clustering text embeddings is a valid proxy for human-defined soft constraints. This is a strong and unverified assumption. The evidence for the methods' effectiveness is therefore only valid for this synthetic task, and the claims of capturing user preferences are not directly supported.

4.  **Inaccuracies and Lack of Clarity in Technical Details**:
    *   The paper incorrectly refers to the hyperparameters `α` and `β` of the `Cosine` method as trainable "parameters" in both the abstract and Section 5.1.
    *   The architecture of the `NQR` model is poorly described and  lacks a key model architecture figure. Table 5 in the appendix is confusing and does not clearly map to the formulation in Equation `(11)`. It is unclear how the self-attention over preferences is used in the final score adjustment.

**Requested Changes:**

1.  **Strengthen Formalism and Clarity**:
    *   Provide a rigorous, formal definition for "hard constraints" and "soft constraints". Clearly delineate why they cannot be handled by extending the FOL query language or KG schema. The current informal definition is insufficient.
    *   Clarify all mathematical notations. For example, in Equation `(2)`, explicitly define what the superscript `i` in `c_j^i` represents.
    *   Add a clear architectural diagram for the NQR model that visualizes all inputs, layers, and outputs, and ensure it aligns with the textual description and equations.

2.  **Improve Technical Justification and Novelty**:
    *   Correct the "parameters" vs. "hyperparameters" terminology.
    *   Revise the related work to position this research within the broader context of Learning to Rank (LTR) and interactive re-ranking. The authors must clearly articulate the unique challenges of their problem that prevent off-the-shelf LTR methods from being a direct solution, thus highlighting the paper's specific contribution beyond applying existing techniques.

3.  **Validate the Experimental Methodology**:
    *   The most significant weakness is the reliance on synthetically generated soft constraints. The authors should conduct a small-scale study to validate whether the clusters generated by their method correspond to meaningful, real-world user preferences. Without this, the paper's main claims are built on an unproven foundation.
    *   The set of baselines should be expanded. Consider including other neural re-ranking models or a simpler MLP baseline to better contextualize the performance of NQR.

4.  **Deepen the Experimental Analysis**:
    *   Provide a deeper analysis of *why* the proposed methods behave as they do. For instance, why does NQR perform better with fewer preferences, while Cosine excels with more? Is this related to overfitting or generalization?

5.  **Minor Revisions**:
    *   Rephrase awkward sentences for better readability (e.g., the sentence starting with "We build on methods...").
    *   Ensure all tables and figures have self-contained captions (e.g., define the `∆` symbol in Table 2).

---

> ### Author Response · Authors · 2026-03-17
>
> We thank the reviewer for the careful reading and constructive feedback, and for recognizing the relevance of the problem and the potential interest of the benchmark for the community. We have revised our submission to reflect these changes (shown in green in the PDF). Please find below our response to the requested changes.
>
> **RC1 - Formalization and clarity.** We agree that the definition of soft constraints should be made more precise. In the revision, we have updated Section 4 and expanded the problem formulation to formally distinguish hard and soft constraints. In particular, we define soft constraints as a predicate $s : \mathcal{V} \rightarrow {0,1}$ that cannot be expressed in first-order logic; we introduce the constrained answer set $\hat{A}_q = {e \in \mathcal{A}_q \mid s(e)}$ and specify the desired ranking properties that preserve hard query answers while prioritizing entities satisfying the soft constraint.
>
> We thank the reviewer for pointing out the need for clarity in the architecture. The table in the Appendix referred to an earlier version of our architecture, whereas our final experiments use a simpler architecture (a 2-layer MLP, as described in the main body of the paper). We have corrected the table in the Appendix and added Figure 2 in Section 5.2 to describe the architecture and connect it with Eq. 11.
>
> **RC2 - Technical justification and novelty.** We now more accurately refer to $\alpha$ and $\beta$ in the abstract, Section 5.1, and the Appendix as hyperparameters.
> Regarding the relation to Learning-to-Rank (LTR), we position our work within that literature and already include an LTR baseline in our experiments. However, the objective differs from standard LTR. Typical LTR methods learn to rank relevant versus non-relevant items from scratch. In contrast, our setting starts from an existing ranking produced by a base QA model for the unconstrained query, and the goal is to incorporate preferences while respecting the structure of that original ranking. We have clarified this distinction in the related work.
>
> **RC3 - Experimental methodology.** We agree that this concern should be addressed explicitly. Our use of embeddings to derive clusters is motivated by prior work (cited in Section 6.1) showing a correlation between embedding structure and human notions of similarity. At the same time, we acknowledge that validation closer to our knowledge graph setting is valuable.
>
> In response, we conducted a validation study comparing embedding-based preference sets with human-curated ones. A human annotator provided partitions for the answer sets of 140 queries (10 per query type). The agreement between embedding-based and curated partitions is 79% accuracy (95% CI [76.8, 81.3]). We further evaluate our models on these curated queries and observe the same overall performance trends as in the main experiments. We have included this validation and qualitative examples of the human rationales in Appendices B and E. We thank the reviewer for suggesting this addition, which strengthens the empirical support for our evaluation setup.
>
> **RC4 - Experimental analysis.** We agree that performance with small preference sets is particularly important in practice. Our plots showing preference set sizes from 1 to 10 allow us to analyze the trade-offs involved, and the comparison between methods is nuanced. Our experiments show that the task-specific training of NQR is beneficial for preference sets of size 4 or smaller, as it prevents large drops in ranking performance (measured with MRR) while capturing preferences accurately (measured with pairwise accuracy). Moreover, NQR achieves higher NDCG@10 in both datasets for preference sets up to size 8, indicating a better balance between these criteria. This is explained by the training objective of NQR (Eq. 12), which explicitly optimizes both aspects, unlike the Cosine method. We have made this discussion explicit in Section 7 under a new Discussion paragraph.
>
> **RC5 - Minor revisions.** We have rephrased the sentence for clarity and updated the captions of Tables 2 and 3 to ensure that they are self-contained.

---

> > ### Comment · Reviewer_2YFJ · 2026-05-06
> >
> > Thank you for the detailed rebuttal and the substantial revisions. The paper is significantly improved: the formalization of soft constraints is now much clearer, the NQR architecture and the role of the Cosine hyperparameters are properly clarified, and most importantly, the new human-validation study (showing 79% agreement between embedding-based and curated preference sets) provides meaningful support for the benchmark construction. I also appreciate the added discussion explaining why NQR performs better with small preference sets while Cosine becomes stronger with more preferences. While some limitations remain—most notably that the evaluation still relies partly on proxy-generated preference sets—I believe the authors have addressed the core concerns to a sufficient extent. Overall, the revised paper now presents a well-motivated task, a practically useful benchmark extension, and lightweight yet effective reranking methods, and I am now leaning positive on acceptance.

---

> ### Comment · Action_Editor_kVdN · 2026-05-02
> **Reviewer 2YF**
>
> Reviewer 2YF: Could you please check author revision and rebuttals and let us know your thoughts?

---

### Review · Reviewer_mV6N · 2026-02-03

**Summary Of Contributions:**

### Summary
This paper tackles a key limitation of First-Order Logic (FOL)–based queries in Knowledge Graph Question Answering (KGQA): their inability to express vague, contextual, or preference-driven requirements, often referred to as soft constraints (e.g., “movies similar to …”). The authors model this setting as an interactive process where users iteratively provide positive and negative example entities to refine their intent.

To address this problem, the paper introduces two lightweight reranking approaches, Cosine and Neural Query Reranker (NQR), which revise the scores produced by a base QA model based on the similarity between candidate answers and the user-provided examples. To evaluate these methods, the authors construct a new benchmark by clustering entity text embeddings, thereby simulating sets of soft preferences.

### Strengths
* **Practical Problem Formulation**: The paper addresses a meaningful gap between rigid logical query formulations and the ambiguous, user-centric intents commonly encountered in real-world KGQA scenarios.
* **Efficiency**: The proposed methods are extremely lightweight, introducing only millisecond-level overhead, which makes them suitable for real-time or interactive systems, an advantage over heavier reranking models.
* **Benchmark Contribution**: The work provides a large-scale dataset for soft-constraint QA derived from established benchmarks such as FB15k-237 and Hetionet, which could be useful for the research community.

### Weaknesses
* **Tautological Evaluation**: The ground truth for soft constraints is generated through embedding-based clustering, while the proposed methods also rely on embedding similarity. This raises the concern that the evaluation may primarily reflect alignment within a single embedding space rather than genuine preference modeling.
* **Unrealistic User Assumptions**: The evaluation emphasizes performance improvements over as many as 10 interaction rounds, which may exceed what real users are willing to tolerate in practice.
* **Weak Baselines**: Comparisons are limited to LightGBM. The absence of Large Language Model (LLM) baselines, which are currently the dominant approaches for handling ambiguity and soft constraints, weakens the assessment of the method’s effectiveness.
* **Simplicity vs. Expressiveness**: Reliance on linear combinations (Cosine) or shallow MLPs over averaged similarities (NQR) may be insufficient to capture more complex, non-linear user preferences (e.g., XOR-like patterns).

**Audience:**

Yes

**Audience Explanation:**

The problem of reconciling structured knowledge graphs with unstructured and vague user intent is highly relevant to researchers in Information Retrieval, the Semantic Web, and Conversational AI. The emphasis on computational efficiency is particularly attractive for practitioners building low-latency systems where large neural models are impractical. Moreover, the release of datasets that extend FB15k-237 and Hetionet with preference signals represents a potentially valuable resource for the community.

**Broader Impact Concerns:**

The paper would benefit from a brief discussion of potential bias amplification. Because the method depends heavily on pre-trained text embeddings to define similarity, any biases present in those embeddings (e.g., gender, racial, or cultural stereotypes associated with entities) may be propagated or amplified during reranking. If a user’s soft constraint aligns with a stereotypical cluster, the system could aggressively filter out diverse yet valid answers. A short acknowledgment of this dependency on embedding quality and bias would strengthen the broader impact discussion.

**Claims And Evidence:**

No

**Claims Explanation:**

Although the experimental results are internally consistent within the authors’ experimental setup, the evidence is not fully convincing due to a potential circularity in the evaluation design (Weakness A).

Specifically, the “ground truth” soft constraints are generated by clustering entity text embeddings, and the proposed methods attempt to satisfy these constraints by computing cosine similarity within an embedding space. In effect, the task is defined in terms of vector-space proximity, and the solution is likewise to recover vector-space proximity. This creates a tautological setup in which the model is rewarded for reconstructing the geometry of the embedding space used for data generation, rather than for genuinely learning or inferring user preferences.

Without validation on human-annotated preference data, or on data generated from a substantially different modality or ontology, it is difficult to conclude that the proposed methods capture soft constraints in a general or user-meaningful sense.

In addition, the strength of the claims is further undermined by the lack of strong baselines (Weakness C). While the paper compares with LightGBM, Large Language Models have become the de facto standard in KGQA for handling semantic ambiguity and few-shot reasoning. Without a comparison to an LLM-based reranker, the evidence for state-of-the-art relevance remains incomplete.

**Requested Changes:**

I recommend the following revisions to strengthen the submission and address the key concerns regarding evaluation design and baseline comparisons

### Critical Changes (Required for Acceptance)
* **Mitigate the Tautology Risk (Weakness A)** The current evaluation uses the same modality, text embeddings, to generate and solve the problem. To demonstrate that the approach is not merely exploiting the geometry of a specific embedding model
  * *Experiment*: Conduct an evaluation where the embeddings used to generate preference clusters differ from those used by the QA model or reranker (e.g., generate clusters with GTE-Qwen embeddings but rerank using BERT or RoBERTa). This would help show that the method generalizes across semantic spaces rather than measuring self-similarity.
  * *Discussion*: Explicitly acknowledge and discuss this limitation in the paper.

* **Include Stronger Baselines (Weakness C)** A comparison limited to LightGBM is insufficient given the current landscape.
  * *Experiment*: Include an LLM-based zero-shot or few-shot reranker (e.g., GPT-3.5/4 or an open-weight model such as LLaMA-3). Even if such models are significantly slower, they would provide a meaningful upper bound. The paper should explicitly quantify the trade-off, for example: “Our method achieves X% of LLM performance while being Y× faster.”

* **Analyze Early-Stage Interactions (Weakness B)** In realistic settings, users are unlikely to provide 10 rounds of feedback.
  * *Analysis*: Figure 3 suggests that for small numbers of interactions (e.g., 1–3), ranking performance (MRR) often degrades relative to the unconstrained baseline. Please analyze why this early-stage “greedy” degradation occurs.
  * *Metrics*: Report results for k=1 and k=3 prominently in tables, as these settings are the most relevant for real-world interactive systems.



### Strengthening Changes (Recommended)
* **Discussion of Linearity (Weakness D)** The Cosine update relies on a linear combination of examples. Please discuss or empirically demonstrate scenarios where this formulation fails. For instance, if a user prefers Action or Romance movies (a bimodal or disjunctive preference), averaging embeddings may land in an unintended region of the space (e.g., Drama), failing to capture the true intent.

* **Qualitative Analysis** Expand the qualitative analysis (e.g., Table 3) to include failure cases, particularly during early interaction steps (small k). This would provide more insights on how and why the model behaves poorly in those regimes.

---

> ### Author Response · Authors · 2026-03-17
>
> We thank the reviewer for the detailed and thoughtful feedback. We appreciate the recognition of the practical importance of the problem, the efficiency of the proposed methods, and the value of the benchmark contribution. We have revised our submission to reflect changes (shown in green in the PDF). Please find next our response to the requested changes.
>
> **RC1 - Mitigating the tautology risk.** We appreciate this concern and agree that it should be addressed explicitly. Importantly, the embeddings used to generate the preference sets do not rely on the same information as the embeddings used by the reranking methods: preference sets are generated from textual embeddings, while reranking methods rely on the graph embeddings (with no access to textual information) of the base QA model. This allows us to generate preference sets that capture vague preferences (i.e. derived from text) and to address a tautology risk. We have added this clarification in Section 6.1 (Datasets-Embedding textual descriptions).
>
> To further investigate the validity of our evaluation for estimating human preferences, we have carried out a validation study comparing embedding-based preference sets with human-curated ones. In this validation, a human annotator provides a partition for the set of answers to 140 queries (10 per query type). The agreement between embedding-based partitions and curated ones is 79% accuracy (95% CI [76.8, 81.3]). We also evaluate our models on these curated queries and observe the same overall performance trends as in our main experiments. We have included this validation and qualitative examples of the human rationales in Appendices B and E. We thank the reviewer for suggesting this addition, which we believe significantly strengthens the empirical support for our evaluation setup.
>
> **RC2 - LLM baselines.** We agree that LLMs are relevant for handling semantic ambiguity. However, we consider them outside the scope of our research for two reasons. The first reason is that we do not assume the base QA model nor the reranking methods have access to textual information. The second reason is that in our target setting thousands of entities may need to be reranked per query. In this setting, applying an LLM for scoring or pairwise comparisons is considerably more expensive than the lightweight methods we study. To confirm this, as part of our validation study we measure the runtime of an LLM when clustering answer sets of up to 20 entities (which is orders of magnitude lower than the approximately 14k entities in FB15k237) for 140 queries, and found that the average runtime per answer set is 3.5 seconds (95% CI [3.3, 3.8]). While this runtime includes network effects, it is orders of magnitude larger than the slower method in our experiments (LightGBM, with a 0.138 sec. overhead) which contain hundreds of thousands of queries, each requiring reranking 14k entities in FB15k237, and 45k entities in Hetionet per query. This is a very important context, which we have now added in Section 7 (under “Reranking with large language models”).
>
> **RC3 - Early-stage interactions.** We agree that performance with small preference sets is important. Our plots already show results for t =1 to 10, allowing analysis of early-stage trade-offs.
>
> Pairwise accuracy captures whether preferred entities are ranked above non-preferred ones, whereas MRR/H@k depend on the absolute positions of correct answers. With few preferences, reranking can correctly separate preferred from non-preferred answers (high pairwise accuracy) but still introduce spurious promotions of non-answers, temporarily lowering MRR. As more preferences are added, the signal becomes more stable, reducing these effects and improving both pairwise accuracy and ranking metrics. We have clarified this behavior in Section 7 with a new discussion paragraph.
>
> **RC4 - Discussion of linearity.** We would like to clarify that our methods do not average embeddings of preferred entities. As shown in Eqs. (10) and (11), the reranking update operates on scores, where similarity signals from positive and negative preferences are added or subtracted. Consequently, if preferences are bimodal (e.g., two distinct clusters), their effects contribute additively rather than being collapsed into a single averaged embedding.
>
> **RC5 - Failure cases.** We agree that it is useful to show failure modes more explicitly. In the revision, we expand the qualitative example in Table 3 to include incorrectly demoted answers, i.e., valid answers that move down in the ranking to the point that they are overtaken by incorrect entities after reranking. This makes the trade-off more transparent by showing not only which preferred answers are promoted, but also which correct answers are harmed and by how much.

---

### Review · Reviewer_Sy7H · 2026-03-03

**Summary Of Contributions:**

This paper addresses the problem of answering queries on knowledge graphs (KGs) with "soft constraints." Unlike traditional methods that rely on strict logical forms, this paper proposes an interactive setting where users provide examples (positive and negative) to refine the query results. The authors introduce two lightweight methods to re-rank the candidate entities: a linear method based on cosine similarity and a neural query reranker. They also construct new benchmarks based on FB15k-237 and Hetionet by clustering textual embeddings to simulate soft constraints. Experiments show that the proposed methods can effectively incorporate user preferences while maintaining the quality of the original ranking.

Strengths:

- The paper identifies an important and practical problem. Handling "soft" or vague constraints is very relevant for real-world search applications and complements existing logical query answering methods.

- The proposed solutions (cosine update and NQR) are computationally efficient. They do not require retraining the heavy QA model and add very little latency to the inference process.

- Since there are no existing datasets for this specific task, the authors propose a reasonable way to generate synthetic preference data using hierarchical clustering of textual descriptions.

Weaknesses:

- The paper compares the proposed methods against a LightGBM baseline. However, since this task is very similar to "relevance feedback" in information retrieval, it would be beneficial to compare it against classic feedback methods. Since the "Cosine" method basically uses the average similarity, comparing it with a simple vector manipulation baseline would highlight the necessity of the proposed formulation.

- The experimental results (Figure 3) show that the simple "Cosine" method often performs better than or comparable to the "NQR" method, especially when the number of interactions increases. NQR requires training an MLP and tuning more hyperparameters, while Cosine is parameter-free (or has very few parameters). The authors should provide more analysis or examples to explain when and why one should choose NQR over the simpler Cosine method. If the neural network does not bring significant gains, the motivation for using it becomes weak.

- The preference data is generated by clustering textual embeddings. Therefore, the "soft constraints" essentially measure the semantic similarity of text. The proposed methods also use embedding similarity to re-rank. This creates a situation where the evaluation might be tautological (i.e., the model works well because it uses the same type of similarity that was used to create the ground truth).

**Audience:**

Yes

**Audience Explanation:**

The paper is related to ML.

**Broader Impact Concerns:**

NA.

**Claims And Evidence:**

No

**Claims Explanation:**

More baselines are necessary to validate the effectiveness of the method.

**Requested Changes:**

Suggestions for improvement:

- Add more baselines and comparisons.

- Add justification for NQR.

- Show the validity of the synthetic data.

Overall, this is a solid paper with a clear motivation and a practical solution. The idea of interactive KGQA with soft constraints is valuable. If the authors can address the concerns regarding the baselines and the data construction, the paper will be significantly improved.

---

> ### Author Response · Authors · 2026-03-16
>
> We thank the reviewer for the positive assessment of the problem formulation, efficiency of the proposed methods, and the benchmark construction. We also appreciate the suggestions regarding baselines, the role of NQR, and the evaluation design. We have revised our submission to reflect changes (shown in green in the PDF). Please find next our response to the requested changes.
>
> **RC1 - Relevance-feedback baseline.** It is true that relevance feedback methods, such as the Rocchio algorithm, are related. However, such methods are based on updating a query vector with a new one as examples are added. In our case, our starting point are the scores of a base QA model for the unconstrained query, for which no query vectors are available that can be updated. The closest methods that we found in the information retrieval literature are learning-to-rank methods, which we already consider in our experiments. We have updated the related work (Section 2) on Information Retrieval and Recommender Systems to emphasize this crucial difference with relevance feedback methods.
>
> **RC2 - Justification for NQR.** We agree that the manuscript should explain more clearly when NQR is preferable to Cosine. Our experiments show that the task-specific training of NQR is beneficial for preference sets of size 4 or smaller, as it prevents large drops in ranking performance (measured with MRR) while capturing preferences accurately (measured with pairwise accuracy). Better performance in such small preference sets is particularly desirable in settings where obtaining them is costly, for example when provided by humans. Moreover, the fact that NQR results in higher NDCG@10 in both datasets with preference sets of size up to 8 indicates that it better balances each of the two criteria measured with MRR and pairwise accuracy. We have expanded on this point in Section 7 in our revised submission.
>
> **RC3 - Validity of the synthetic data.** We appreciate this concern and agree that it should be addressed explicitly. Importantly, the embeddings used to generate the preference sets do not rely on the same information as the embeddings used by the reranking methods: preference sets are generated from textual embeddings, while reranking methods rely on the graph embeddings (with no access to textual information) of the base QA model. This allows us to generate preference sets that capture vague preferences (i.e. derived from text) and to address a tautology risk. We have added this clarification in Section 6.1 (Datasets-Embedding textual descriptions).
>
> To further investigate the validity of our evaluation for estimating human preferences, we have added a validation study comparing embedding-based preference sets with human-curated ones. In this validation, a human annotator provides a partition for the set of answers to 140 queries (10 per query type). The agreement between embedding-based partitions and curated ones is 79% accuracy (95% CI [76.8, 81.3]). We also evaluate our models on these curated queries and observe the same overall performance trends as in our main experiments. We have included this validation and include qualitative examples of the human rationales in Appendices B and E. We thank the reviewer for suggesting this addition, which we believe significantly strengthens the empirical support for our evaluation setup.

---

### Decision · Action_Editor_kVdN · 2026-05-08

**Recommendation:** Accept as is

**Audience:**

Yes

**Audience Explanation:**

The topic is of interests to the KG community.

**Claims And Evidence:**

Yes

**Claims Explanation:**

All reviewers are satisfied with the revision and rebuttals, which I agree.